# Low-Cost Sensor Node for Air Quality Monitoring: Field Tests and Validation of Particulate Matter Measurements

**DOI:** 10.3390/s23020794

**Published:** 2023-01-10

**Authors:** Ueli Schilt, Braulio Barahona, Roger Buck, Patrick Meyer, Prince Kappani, Yannis Möckli, Markus Meyer, Philipp Schuetz

**Affiliations:** 1School of Engineering and Architecture, Lucerne University of Applied Sciences and Arts, CH-6048 Horw, Switzerland; 2EQUANS Services AG, CH-8050 Zürich, Switzerland

**Keywords:** air quality monitoring, particulate matter, sensor validation, low-cost, mobile sensor nodes

## Abstract

Air pollution is still a major public health issue, which makes monitoring air quality a necessity. Mobile, low-cost air quality measurement devices can potentially deliver more coherent data for a region or municipality than stationary measurement stations are capable of due to their improved spatial coverage. In this study, air quality measurements obtained during field tests of our low-cost air quality sensor node (sensor-box) are presented and compared to measurements from the regional air quality monitoring network. The sensor-box can acquire geo-tagged measurements of several important pollutants, as well as other environmental quantities such as light and sound. The field test consists of sensor-boxes mounted on utility vehicles operated by municipalities located in Central Switzerland. Validation is performed against a measurement station that is part of the air quality monitoring network of Central Switzerland. Often not discussed in similar studies, this study tests and discusses several data filtering methods for the removal of outliers and unfeasible values prior to further analysis. The results show a coherent measurement pattern during the field tests and good agreement to the reference station during the side-by-side validation test.

## 1. Introduction

Air pollution continues to be a concern as short- and long-term exposure to classical pollutants pose short- and long-term negative effects on human health. A recent study conducted by Juginović et al. [1] shows that, even though levels of air pollution have decreased since 1990 in Europe, it still remains a major public health issue. The recent WHO global air quality guideline recommends setting interim targets and progressing towards lower maximum levels of particulate matter (e.g., PM_2.5_ and PM_10_), ozone, nitrogen, sulfur dioxide (SO${}_{2}$), and carbon monoxide [2]. Switzerland has shown success in controlling air pollution [3], for example, in the case of SO_2_. However, PM is still a concern. Recently, Chen et al. [4] and Rodopoulou et al. [5] conducted fine particle exposure assessment studies in Europe and reported potentially increased mortality given the exposure to several compounds that are found in dust particles. For example, particles of vanadium, chosen as an indicator of petroleum combustion in Chen et al. [4], were shown to increase health risks. Swiss regulatory limit values for average annual particulate matter pollution levels are 20 μg/m^3^ and 10 μg/m^3^ for PM${}_{10}$ and PM${}_{2.5}$, respectively. The daily average limit value for PM_10_ is 50 μg/m^3^ [6]. Recent WHO guidelines are even stricter, recommending yearly average values of 15 μg/m^3^ and 5 μg/m^3^ for PM${}_{10}$ and PM${}_{2.5}$ and daily average values of 45 μg/m^3^ and 15 μg/m^3^ for PM${}_{10}$ and PM${}_{2.5}$, respectively [2].

The decarbonization of our energy consumption calls for combustion-based sources of particulate matter, such as those from burning oil, to be phased out. However, non-exhaust sources of particulate matter, such as those from vehicle’s braking systems and wear of tires, might not be as easily eliminated if people simply switch to electrical vehicles [7]. Continuous monitoring of the air is therefore very important to steer towards significant health improvements globally. Low-cost sensors present a possibility to increase the density of measurements in a given region in a cost-effective way. For this purpose, many different sensors are available on the market. These vary not only in working principle and performance, but also in price [8,9,10].

The standard reference method for measuring particle mass concentration and size distribution in ambient air is the gravimetric method, which uses filters to collect the different particle sizes. Weighing of the filters prior to and after the sample collection allows one to determine the particle mass concentration. Even though this method is found to be accurate, sensitive, and robust, it has some disadvantages. Due to the integrative method, results are only available with a time-delay (usually in the range of days) and not in real-time [11,12]. Therefore, other methods can be used to obtain real-time measurements and higher time resolution. Direct-reading, low-cost sensors typically can be categorized into one of two working-principles: optical particle counters (OPC) and photometers. Both types are based on the light-scattering principle, where the aerosol particles are passed through a light beam. OPC sensor types measure the intensity of the light scattered by each single particle and calculate the size distribution of the particles thereof. Photometers, however, measure the total amount of light scattered by the aerosol particles present in the sensor and calculate the particle concentration in the air [13,14].

Several studies have looked at validating and calibrating particulate matter (PM_2.5_, PM_10_) measured with low-cost optical sensors. The results of these studies are varied. An overview is presented in Table 1, where information about the experimental setup, as well as results are presented. While the PM sensor used in the study presented here is of the OPC type, studies using both measuring principles, OPC and photometry, have been looked at. Most studies discussed below have been carried out with side-by-side testing, meaning the low-cost sensor nodes are located directly adjacent to the reference station. One study conducted by Penza et al. [15] in Bari, Italy, employed a network of 11 sensor nodes, including one mobile node. These results, however, were not compared side-by-side with a reference station, but with the closest air quality monitoring station. An analysis of three sensor nodes showed good agreement with the monitoring station data (mean absolute error of 5.6 μg/m^3^). A side-by-side study conducted in Aveiro, Portugal by Borrego et al. [16], resulted in relatively low correlations ($r^{2}$: 0.13–0.36 for PM${}_{10}$ and 0.07–0.27 for PM${}_{2.5}$). The measurements for this study were taken at an urban traffic location in the city center. In a study conducted by Castell et al. [17] with 24 identical commercial sensors in Oslo, it was shown that the performance varies from unit to unit. The calibration was conducted with linear regression in this case. As a conclusion, it is suggested that the calibration of the nodes should be carried out in an environment similar to where they will be deployed. Other recent studies were carried out in Seoul, where sensor nodes were co-located with reference monitoring stations: Lee et al. [18] applied a combined (linear and non-linear) calibration method called SMART (Segmented Model and Residual Treatment) to the PM data, while Park et al. [19] developed a calibration model called HybridLSTM, combining a deep neural network and a long short-term memory neural network in order to improve the correlation. During a field test conducted in Helsinki, measurements of PM_2.5_ concentrations were performed using portable air quality sensors [20]. Indoor as well as outdoor measurements were performed. It was found that all measurements were consistent through validation among themselves. The measurements also showed good agreement with a nearby reference station. Arroyo et al. [21] carried out a study in Badajoz, Spain, where two portable devices for outdoor air quality measurements were placed adjacent to a reference station located in a traffic hot-spot. The applied calibration methods were simple linear regression, multiple linear regression, and a multilayer perceptron artificial neural network. Depending on the selected calibration method, the PM sensors showed a good performance when compared to the reference station. Another study carried out at two different locations in Italy—Ispra (North Italy) and Brindisi (South Italy)—evaluated the accuracy of PM_10_ measurements acquired with low-cost sensor nodes [22]. The portable sensor nodes were placed side-by-side with reference stations for a duration of approximately five months with a sampling rate of one sample per minute. Mean and maximum error (compared to reference station data) were calculated as 9.0 μg/m^3^ and 41.7 μg/m^3^, respectively. This result was judged as a good agreement. In Motlagh et al. [23], the opportunities and challenges of a large-scale deployment of air quality sensors are discussed, including use cases, as well as key requirements. The results of a testbed deployment in Helsinki are presented, where sensors of different types have been placed in three different environments (industry, residential, and mixed). The mobile sensors were calibrated with data from fixed reference stations located in the vicinity of the sensors.

Most of the studies presented above contain one of the two situations: either a side-by-side comparison of stationary sensor nodes, or an evaluation of portable sensor nodes, where the closest available reference station is used for calibration. Our analyses presented in this paper aim at evaluating the suitability and reliability of air quality data acquired with mobile low-cost sensor nodes of the OPC type. Therefore, we develop a low-cost sensor node (sensor-box) that can be mounted on a vehicle and perform field tests with utility vehicles of municipalities in Central Switzerland. Our sensor-box measures air quality, temperature, humidity, ambient sound, and ambient light. Side-by-side comparisons against reference stations let us validate our measurements and design raw data filters. Here, we present the performance of our temperature and PM_10_ measurements in field tests and a validation with a reference station operated by the regional air quality monitoring network.

In Section 2, the methods and equipment used for the data acquisition and processing are described. The setup for the validation measurement and the field tests is presented. Furthermore, a short overview of historic air quality monitoring data from Central Switzerland is given. Section 3 presents the results obtained from both the validation measurements as well as the field test campaign. A filtering method for processing the raw data is introduced, and the obtained measurements are compared to data from reference stations. Finally, in Section 4, conclusions are drawn from the presented study and possible future work is suggested.

## 2. Materials and Methods

### 2.1. Low-Cost Sensor Node

In the study presented in this article, we develop a low-cost sensor node (sensor-box) to measure ambient air quality (NO${}_{2}$, O${}_{3}$, TVOC, CO${}_{2}$eq, PM${}_{1}$, PM${}_{2.5}$, PM${}_{10}$), temperature, humidity, ambient sound, and ambient light. It can be mounted on top of a utility vehicle and records geo-tagged measurements. The idea is to acquire environmental data as the vehicle is operated by personnel of a municipality to perform tasks such as garbage pick up and gardening. This operation creates a data set of spatially distributed measurements within a community.

Our sensor-box prototype is comprised of several low-cost sensing devices, which are housed in a water-resistant plastic enclosure. The sensor-box can be mounted on top of a vehicle using magnets, therefore acting as a mobile air quality measurement unit. An overview of the sensor-box layout can be seen in Figure 1. Two microcontrollers (FiPy and ESP32) (A) are used for collection of data from the sensors, intermittent storage, data transmission, and power management. The reason two microcontrollers are used instead of one is that the processing of sound measurements is computationally very intensive. While sound data are processing, no other signals can be processed. Therefore, an additional microcontroller reduces computation time. Data can be transmitted via low-power wide-area networks (LoRa), local area networks (WiFi), and broadband (LTE). In this case, we focus on the demonstration of LTE functionality. The LTE antenna (B) used by the FiPy microcontroller is also shown in Figure 1. Additional components include: GPS antenna (C); DC/DC converter (D) to step down the car battery voltage (i.e., 12 V, or 24 V) to 5 V; TSL2691 sensor (E) to measure light and IR data; electrochemical sensors OX-A431 and NO2-A43F from Alphasense (F) to measure O${}_{3}$ and NO${}_{2}$; three CMA-4544PF-W microphones (G); and SHT35 and SGP30 sensors from Sensirion (H) to measure temperature and humidity and TVOC and CO${}_{2}$eq, respectively. The focus in this study is on the performance of the PM3015SN sensor from Cubic (I) to measure particulate matter PM${}_{10}$ concentrations [24]. Table 2 shows the most important specifications of the PM sensor, including the accuracy of the measurement. The air circulation of the sensor box is enhanced by the fan of the PM-sensor and an externally mounted snorkel. Additionally, the ground plate of the box has several holes.

Once a box is mounted on a vehicle by magnets and connected to the power supply it automatically starts to record data. The measurements are taken in cycles, as shown in the software flow chart in Figure 2. When the sensor-box is connected to a power source, the start-up (boot process) is automatically initiated. The SD card, which contains software libraries and sufficient space for data storage, is connected to the micro-controllers. The libraries are then loaded and a box-specific ID identifies the sensor-box. As a next step, the sensors and the GPS modules are initialized, meaning the GPS is searching for satellite signals. If, after several attempts, a GPS signal cannot be found, the boot process restarts. Start-up of the sensor-box is completed once the GPS signal has been acquired. The measurement cycles will then start: each sensor takes a measurement, and the time and geo-location are recorded as well. The system then proceeds to store the data locally on the SD card, before the LTE module tries to establish a connection to the network. If a connection can be established, the data are sent to the server for storage. If the LTE connection cannot be established, the data are stored locally on the SD card and uploaded later, when a connection can be established. A cycle of measurements, data storage, and transmission is carried out approximately every 30 s. The PM sensor requires a short time (≤8 s) for start-up before it can take measurements (time to first reading). The boot process of the cycle shown in Figure 2 takes long enough for the PM sensor to ensure such a start-up time.

In the study presented in the subsequent sections, a total of 15 sensor-boxes have been deployed. Each sensor-box is labeled with a number ID from 1 to 15.

### 2.2. Sensor Node Cost

The sensor node presented in this study is considered low-cost in comparison to more high-grade air quality measurement devices. The price range of different types of air quality monitoring stations is discussed in Motlagh et al. [23]. There, it is mentioned that a professional-grade measurements station with high-precision sensing instruments can reach costs in the range of hundreds of thousands of dollars. In comparison, low-cost portable monitoring stations typically do not exceed costs of USD 2500. Streuber et al. [25] uses two types of low-cost sensing units for comparison in a laboratory setting: the in-house developed air-monitoring platform GeoAir2, which is based on a Sensirion SPS30 PM sensor, and an Alphasense OPC-N3 PM sensing unit. The GeoAir2 comes at a cost of USD 250–350, depending on equipment, while the Alphasense OPC-N3 is mentioned to cost USD 500. Bean [26] evaluated four different brands of low-cost particulate matter sensors during a measurement campaign. It is also mentioned, that all four sensors cost less than USD 300 each. The cost of air quality sensors is also mentioned in Castell et al. [17], stating that the price for fixed-site monitoring stations with certified reference instruments ranges from EUR 5000 to 30,000, whereas the cost for commercial low-cost sensor nodes varies between EUR 500 and 5000.

The cost of the sensor-box used in this study lies between EUR 600 and 1000 for the complete sensor node. The PM sensing device costs in the range of EUR 40–50. Therefore, it falls into the category of low-cost sensor nodes.

### 2.3. Validation Setup

In order to validate the sensor-box measurements, the sensor-boxes are set up to have nearly the same environment as the in-luft measurement station. This way, the influence of a changing environment as experienced on mobile sensor-boxes can be eliminated. Therefore, a comparison to a reference instrument was performed. In this study, a set of three boxes with the IDs 1, 2, and 7 were considered. The sensor-boxes were placed side-by-side with a reference instrument part of the air quality monitoring network in-luft (Section 2.5). This validation campaign was held from mid October 2021 to the start of January 2022 next to an in-luft station located in Stans. During this period, three sensor boxes were mounted on the cabinet of the reference station as shown in Figure 3. Two of the three sensor-boxes were mounted on top of the gray plastic box. In the following, this sensor-box setup is annotated as “normal”. The third sensor-box was placed inside the gray plastic box. This third sensor-box was left without a cover in order to have similar environmental conditions as the reference station, since the closed sensor boxes have limited air circulation. To ensure improved air circulation in the gray box, an air fan was mounted.

The specifications of the measurement device Fidas200 used in-luft are shown in Table 3. It can be observed that the Fidas200 device is a more advanced measurement device than the low-cost PM3015SN employed in the low-cost sensor-box. The Fidas200 is based on the OPC measurement method, working with a volumetric air flow of approximately 0.3 m${}^{3}/$min [27]. In addition, the device is equipped with a heating device, reducing the humidity of the incoming air before measuring its PM concentration. This is important for optical measuring devices, as humidity increases the particle diameters, therefore changing the refractive properties, which in turn results in an increased sensor output signal [13,28]. The mass concentration would therefore be overestimated.

### 2.4. Field Tests Setup

The sensor-boxes are mounted on the roof of a municipal utility vehicle using four 89 N adhesive force magnets, provided the roof is magnetic. The four magnets are directly attached to the plastic enclosure, as can be seen in Figure 1J. In order to ensure that the magnetic forces are sufficient and a loss of the sensor-box during vehicle operation can be ruled out, the adhesive forces of the magnets when mounted to the sensor-box were tested in the lab. The GPS antenna unit is also attached to the roof with a magnetic surface. The power for the box is directly provided by the car battery (12 V or 24 V, depending on the vehicle) by routing a cable from the battery to the box. Figure 4 shows the sensor-box mounted on the roof of a municipal utility vehicle.

During the pilot-phase of the measurement campaign, 14 communities agreed to have sensor-boxes mounted on their vehicles. One sensor-box was mounted per pilot (i.e., community). The first pilots started operating at the end of April 2021, and the pilot phase ended in April 2022. Some of the pilots were decommissioned earlier, such that data from 4 months to 1 year were gathered with the corresponding pilots. Table 4 shows an overview of the pilots and the respective campaign duration. With this time-span all the seasonal effects such as temperature, rainfall, heating season, and summer season are covered in the collected data. During the campaign, the system was continuously improved and adapted to fix common bugs on the hardware and software sides.

### 2.5. Air Quality Monitoring Data from Central Switzerland

Monitoring stations are operated by national and cantonal environmental offices in order to fulfill regulations such as those established by the Swiss Federal Act on the Protection of the Environment and by the Ordinance on Air Pollution Control. In the case of Central Switzerland, six cantons operate a network of fixed monitoring stations (in-luft) that measure air quality [29]. There are currently ten locations where in-luft measurements of concentrations of nitrogen-oxides (NO${}_{x}$), particle matter (PM${}_{10}$, PM${}_{2.5}$, PM${}_{1}$, and soot), ozone (O${}_{3}$), ammonia, and volatile organic compounds (VOC) are taken. Here, we use part of these public data to validate our sensor-box PM_10_ measurements and to verify the measurements during the pilot tests.

According to the in-luft measurements in the year 2020, pollution levels for particulate matter PM_10_ and PM_2.5_ complied with regulations in every location. Higher concentrations were observed at sites with heavy traffic in larger cities. Daily mean limit values were also complied with at each location. However, large-scale phenomena, such as the arrival of Saharan dust, caused larger concentrations at the end of March. Elevated concentrations also usually occur during the winter months, driven by temperature inversions and poor mixing of air masses in urban streets. In rural and higher-altitude areas, particulate matter concentrations were the lowest [3].

At a national level, data from the Swiss Federal Office for the Environment (BAFU) show that between 1986 and 2019, PM_10_ pollution levels decreased by 60%. The influence of the reduced economic activity due to the COVID-19 pandemic may be observed in these measurements. BAFU’s monthly report from June 2022 shows that hourly and daily values are occasionally higher than desired [30,31]. However, as well the regional in-luft data, yearly pollution levels from July 2021 to June 2022 are below Swiss regulatory limit values. Nevertheless, given their impact on human health, fine and ultra-fine particulate matter pollution (such as PM_2.5_, PM_1_, and soot) should be further reduced.

### 2.6. Quality Control of Raw Air Quality Data

Research work that uses low-cost sensors for measuring particulate matter pollution does not typically discuss the processing of raw sensor data that might be necessary to apply before performing calibration against a reference station. In recent work carried out by Cummings et al. [32], the top and bottom 0.5% of measurements are removed to account for outliers, and data lacking geotags are also removed. However, emissions from nearby vehicles are not filtered out in an attempt to retain insights regarding traffic density and pedestrian’s exposure to high pollutant concentrations. Earlier work, such as that carried out by Borrego et al. [16], describes approaches used to use uncertainty metrics to meet European guidelines for data quality. Technical documents describe the quality control processes applied in practice [3,33,34,35,36]. These include automated checks and those performed by analysts. LaGuardia and Hafner [33] describe two of such steps for data quality control, starting first automated checks on ranges, rate of change, sticking values, and drifts. All of these are flagged and can be edited at a later stage by an analyst via a web interface that allows for the comparison of hourly data values to nearby stations and batch editing of data to apply bias and scaling corrections. Generic aspects of the measurement procedures and data quality assurance steps are also described in Zentralschweizer Umweltfachstellen [3]. Data are collected continuously in the measuring stations, and these raw values are aggregated in time and consolidated in a database where the following plausibility checks are performed: violation of threshold values, jumps, identical values, and certain device states are imputed with statistical methods. In addition to these automated quality checks, calibrations are also performed regularly as described in Zentralschweizer Umweltfachstellen [3]. Particularly, PM${}_{10}$ and PM${}_{2.5}$ measurements are calibrated with gravimetric fine dust measurements.

Part of this study is the pre-processing of the raw sensor data before further analysis is performed on the data. Therefore, the last stage of our pre-processing pipeline prior to validation of the sensor-box is removing statistical outliers. Several approaches were tested aiming at removing the minimum amount of data in order to keep extreme values but remove statistical or physically unfeasible values. In order to select the most suitable filtering method for the mobile pilots, seven filtering methods were tested on the data sets gathered during this validation. Among others, the methods described in Leys et al. [37] and Kulanuwat et al. [38] were also tested. An overview and description of the seven filtering methods is given in Table A1 in the Appendix A. Filters 1 and 2 are applied to the complete data sets, while Filters 3–7 are applied to the data using a sliding window with a given window size. Symmetrically around each data point of the data set, an upper and lower band for the window is calculated. The data point is then evaluated against the thresholds: if it falls outside the upper or lower threshold, it is considered to be an outlier and removed. For all the filtering methods with a moving window, Filter 1 (fixed upper limit) is applied first before applying the moving window filter as this removes points that are known to be non-physical, such as, e.g., a constant value of 1000 μg/m${}^{3}$ over several hours.

When comparing the hourly sensor-box data to the hourly in-luft station data, the suitability of each method is analyzed using time-series plots, scatter plots, histogram plots, Pearson correlation coefficient $R_{P}$, and Spearman’s rank correlation coefficient $R_{S}$. The results of this pre-processing step are described in Section 3.1.

### 2.7. Data Analysis and Validation Methods

The data analysis is carried out in two steps: first a suitable filtering method for the raw data is selected based on the validation measurements described in Section 2.6. Subsequently, the selected filter is applied to the raw data set prior to all further analyses. In order to validate the sensor-box PM data, it is compared to the reference data obtained by the Fidas200 air quality station. For this purpose, the correlation between reference data and sensor-box data is calculated using Pearson correlation coefficient $R_{P}$ and Spearman’s rank correlation coefficient $R_{S}$. Furthermore, Mean Absolute Error (MAE), Root-Mean-Squared Error (RMSE), Slope, Intercept, and Sensor bias are calculated for each sensor-box. Sensor bias is calculated based on Mean Percentage Error, using the following equation:(1)$$Sensor\;bias=\frac{1}{n}\sum_{i=1}^{n}\frac{C_{PM10,sensorbox,i}-C_{PM10,inluft,i}}{C_{PM10,inluft,i}}*100\%$$
where $C_{PM10}$ is the measured PM${}_{10}$ concentration at time *i* measured by either the sensor-box or the in-luft station. A similar method has been used in Streuber et al. [25]. With sufficient agreement between reference data and sensor-box validation measurement data, the mobile sensor-box data acquired during the field study are then also analyzed using the same metrics. In addition to the statistical methods mentioned above, which are applied to each individual sensor-box, the low-cost sensors are statistically analyzed against each other by computing analytical metrics from the resulting metrics calculated previously: mean, minimum, maximum, standard deviation (SD), variance, and coefficient of variation (CV) are applied to the resulting data series of $R_{P}$, $R_{S}$, slope, intercept, sensor bias, MAE, and RMSE. This provides an insight about the precision of the low-cost sensor model. The CV for each statistical metric is calculated as follows:(2)$$CV=\frac{SD_{m}}{\bar{m}}*100\%$$
where $SD_{m}$ is the standard deviation and $\bar{m}$ is the mean value of the respective statistical metric (e.g., $R_{P}$) across all sensor-box data sets.

For the analysis of the field study data, the sensor-box data are compared to a nearby reference station. Apart from the described filtering method, no further sensor calibration is applied to the data. The sensor-box data, which are acquired in approximately 30 s intervals, are converted to hourly mean values for comparison with the reference station data. This is due to the fact, that the highest available resolution of the reference data is hourly.

## 3. Results and Discussion

### 3.1. Validation with Reference Station

Three sensor-boxes were placed right next to the in-luft station in Stans, as described in Section 2.3. Measurements were recorded over approximately 2.5 months. Table 5 shows an overview of the validation measurement campaign. The goal of this validation campaign is to compare the data quality of the low-cost sensor-box measurements to the high-quality in-luft measurements and derive pre-processing algorithms that account for outliers. Thus, a filtering method to remove outliers from sensor-box data is developed and evaluated. This filter can then later be applied to the mobile pilot measurements in order to improve the data quality, without losing information about extreme values.

Python libraries were used to develop scripts for data evaluation and manipulation.

The in-luft data are available as hourly mean values. Therefore, the sensor-box data are converted to hourly mean values in order to carry out a comparison. Prior to converting the sensor-box data, however, a filtering method for outlier removal is applied to the raw data set. The evaluation of seven filtering methods is described in Section 2.6, and detailed results of the different methods can be found in tables in the Appendix A. The resulting correlation coefficients, as well as the number of data points removed for the analysis of the filtering methods without sliding window (no filter vs. Filters 1 and 2) can be found in Table A2. The results of the filtering methods with sliding window (Filters 3–7) are presented in Table A3 for a window size of 1000 data points and in Table A4 for a window size of 20,000 data points. Window sizes from 100 to 20,000 data points were evaluated. Figure 5 shows the evaluation of the different filtering methods at window sizes 100 and 20,000, as well as the filtering methods with fixed window (complete data set) for data recorded with sensor-box 2.

Based on an evaluation of the results of all seven filtering methods, Filter 2 is chosen for further processing of the data. This method removes all data larger than the specified percentile from the raw sensor-box data. A value of 99.0% percentile is chosen in this case. The evaluation of the filtering methods considers the resulting correlations between sensor-box and in-luft station data, as well as the amount of removed data for each method. A good balance between the two metrics is required. Looking at the graph shown in Figure 5, it can be seen that there are several filtering methods yielding a higher Pearson correlation than Filter 2. However, the increase in $R_{P}$ is accompanied by a much larger percentage of removed data (e.g., Filters 4 and 7 at window size 20,000). Removing too much data poses the risk of losing physically relevant phenomena. Therefore, Filter 2, with a selected percentile of 99.0% provides the best balance between the two metrics.

Table 6 shows the results of the statistical analysis of the three sensor-boxes used for validation. When applying Filter 2 (fixed percentile) with a 99.00 percentile to the sensor-box data, the resulting Pearson correlation coefficients are 0.74, 0.72, and 0.82 for sensor-boxes 1, 2, and 7, respectively. Looking at bias, it can be seen that two sensor-boxes (ID 1 and 2) overestimate the PM concentration, while one sensor-box (ID 7) underestimates the PM concentration. All three slopes are larger than 1, while sensor-box 7 is very close to 1. Figure 6 shows the comparison between the sensor-box PM${}_{10}$ data of box 7 with the in-luft data in a time-series graph, as well as in a scatter plot. A good correlation between the two data sets is observed.

Figure 7a–c show the distribution of the PM${}_{10}$ data of sensor-boxes 1, 2, and 7 in a histogram. For all three pilots the distribution is similar: the largest share of data points falls into the range of 0–10 μg/m${}^{3}$, and the second largest share falls in the range of 10–20 μg/m${}^{3}$, with the number of data points decreasing with increasing PM${}_{10}$ concentration.

### 3.2. Influence of Ambient Conditions on PM${}_{10}$ Measurements

In addition to the comparison with the in-luft measurements, the influence of temperature and humidity on the sensor-box measurements was examined. These results can then be compared with findings reported in literature in order to validate the dependency of recorded PM concentration with humidity and temperature. For this purpose, the temperature and humidity recorded with sensors located in the same sensor-box were used. Additionally, PM${}_{10}$ measurements from the in-luft station were compared to sensor-box measurements to analyze the impact of humidity. Information about the sensors can be found in Section 2.1. Hourly mean data from boxes 1, 2, and 7 were looked at. For all three boxes, the following patterns emerged:

**Temperature**—High PM${}_{10}$ concentrations only emerged at lower temperatures. The reverse, however, is not the case: low PM${}_{10}$ concentrations are also found at low temperatures. Figure 8a shows a scatter plot of hourly mean temperature and PM${}_{10}$ concentrations for sensor-box 1. As an example, all hourly mean PM${}_{10}$ values of 40 μg/m${}^{3}$ or higher were recorded at an hourly mean temperature below 10 ${}^{\circ }$C. Figure 8b shows the distribution of the PM${}_{10}$ measurements across the different temperature levels.

**Humidity**—For the sensor-box readings, high PM${}_{10}$ concentrations only emerged at higher relative humidity. The reverse, however, is not the case: low PM${}_{10}$ concentrations are also found at high relative humidity. Figure 9a shows a scatter plot of hourly mean humidity and PM${}_{10}$ concentrations measured with sensor-box 1, as well as in-luft measurements. As an example, all hourly mean PM${}_{10}$ values of 40 μg/m${}^{3}$ or higher measured with the sensor-box were recorded at an hourly mean relative humidity above 75%. The in-luft measurements, however, do not show such a dependency on humidity: the hourly mean values of PM${}_{10}$ never exceed concentrations of 30 μg/m${}^{3}$ in the same time period. Figure 9b shows the evolution of PM${}_{10}$ measurements from both the sensor-box and the in-luft station in relation to the measured humidity between 4 December 2021 and 24 December 2021. Here, it can be observed that, while there are periods where both measurements are in good agreement (e.g., from 4 December to 12 December), there are periods where the sensor-box measurements far exceed the in-luft measurements (e.g., period around 15 December). It can further be seen that these high PM${}_{10}$ values only occur during periods of high humidity.

The above observations are consistent with other results reported in literature. Hernandez et al. [39] carried out a study in Auckland, New Zealand, where meteorological conditions and PM concentrations were monitored over an eight week period. A negative correlation between temperature and PM${}_{10}$ concentration and a positive correlation between humidity and PM${}_{10}$ concentration were reported. In addition, it was also found that PM${}_{10}$ levels sometimes remained low despite an increase in humidity. Jayaratne et al. [40] examined the influence of humidity on the measurements of PM concentrations recorded with a low-cost sensor in Brisbane, Australia. The sensors showed a steady increase in PM concentrations at high humidity levels above 75%. In some instances, the PM concentration decreased even at high humidity levels, which was the case in the presence of rain. Ramasamy Jayamurugan and Chockalingam [41] analyzed the influence of temperature and relative humidity on PM concentrations in North Chennai, India, during different seasons. PM levels showed a positive correlation with temperature for all seasons except one, and negative correlations were found between relative humidity and PM concentrations for all seasons.

The influence of high humidity levels on particulate matter measurements is well-described in the literature. Alfano et al. [14] mentions that humidity is a relevant environmental parameter and that keeping relative humidity low will avoid the rapid degradation of the accuracy of low-cost sensor modules. That study also mentions how high levels of humidity can result in possible coalescent phenomena, which makes the particle size appear larger and therefore distorts the concentration measurements. This effect is also described in Lanki et al. [28] and Santi et al. [13]. Some of the differences between sensor-box measurements and in-luft measurements observed in Figure 9a,b could be explained by the fact that the in-luft measurement unit (Fidas200) is equipped with a heating device, as described in Section 2.3. Therefore, a distortion of measured particle size and concentration due to humidity is avoided.

Several studies found in literature show similar results. Crilley et al. [42] compared low-cost OPC sensors placed in an urban setting to reference measurements. There it was also observed that lower relative humidity resulted in better agreement between low-cost sensor measurements and reference measurements. Measurements taken at high relative humidity (i.e., >85%) showed an exponential increase in OPC PM concentration readings in relation to the reference measurements with increasing humidity levels. Streuber et al. [25] evaluated two types of low-cost particulate matter sensors in a laboratory setting, using high and low mass concentrations. It was also observed that the effect of hygroscopic growth due to increased relative humidity lead to a increased overestimation of the particle concentration. Wang et al. [43] evaluated the performance of three low-cost PM sensors based on the light-scattering principle under laboratory conditions. Among others, the influence of temperature and humidity on the sensor performance was examined. It was shown that temperature had a negligible effect on the sensor measurement, while relative humidity affected the sensor performance significantly. Particle mass was overestimated due to altered absorption properties. Bai et al. [44] conducted a long-term field experiment where the capabilities of low-cost PM sensors were evaluated. They were co-located with a reference measurement device. Calibration was carried out using linear and non-linear regression, as well as an artificial neural network. It is reported that high relative humidity (i.e., >75%) leads to higher errors in measured PM concentration. Temperature, on the other hand, was found to have a negligible effect on sensor performance. A study conducted by Di Antonio et al. [45] also showed an overestimation of measured PM concentrations by low-cost sensing devices (OPC) at high humidity levels. In this case, the performance of the OPC device was improved by applying a particle-size distribution-based correction algorithm. Similarly, Zheng et al. [46] reported major influences of high humidity levels (>70%) on low-cost PM sensors and applied corrections using empirical nonlinear equations.

As consistently shown in the above-mentioned studies, it can be expected that the low-cost PM sensor measurements will produce overestimated values of PM concentrations when exposed to high relative humidity.

### 3.3. Measurements with Mobile Sensor Nodes

Field-tests were carried out with mobile sensor-boxes mounted on several vehicles in the region of Central Switzerland. The test-setup is described in Section 2.4. Data were recorded between April 2021 and April 2022. For the analysis described in this section, only data recorded until the end of December 2021 are considered. Figure 10 and Figure 11 show the time-series graph of hourly aggregated data for two selected months—July and December. Only pilots containing at least 100 mean hourly data points per month are represented on the graphs.

In Figure 10 (July), it can be seen that some pilots delivered PM${}_{10}$ values of similar magnitude (e.g., AEW, Cham, Emmenbruecke, Hergiswil, Horw, Kriens, Olten), while other pilots differ in magnitude (e.g., Lostorf, Malters, Stansstad). Similarly, this can be observed for the month of December in Figure 11.

The acquired data of the mobile sensor nodes are evaluated against data from nearby in-luft air quality stations where such stations are available. The procedure for the comparison is as follows: first, a fixed percentile filter (Filter 2 acc. Table A1) is applied to the raw sensor-box data (99.0 percentile). Then, the raw sensor-box data are converted to mean hourly values before being compared against the hourly in-luft data.

Throughout the measurement campaign, it was sometimes required to exchange a sensor-box at a specific pilot location due to hardware problems. Therefore, in some cases, multiple sensor-boxes were used sequentially at the same pilot location. At any given point in time, no more than one sensor-box was deployed at a specific pilot location. The evaluation is carried out for each box individually so that each data set only contains data obtained with the same hardware. Data are only evaluated if there are sufficient data available for several consecutive days. Considering the aforementioned restrictions, 21 usable data sets resulted from the measurement campaign between 1 May 2021 and 31 December 2021. The 21 data sets are labeled with letters from (A) to (U), as shown in Table 7. The table further shows the pilot location, sensor-box ID, the in-luft station used for reference, the number of available mean hourly data points, as well as the distance between in-luft station and pilot, rounded to the nearest integer kilometer value. While the position of the in-luft station is fixed, for the location of the mobile pilot, the approximate center of its area of movement is used. The amount of data collected differs widely between the different sensor-boxes. Sensor-box 2 in Malters only has 75 hourly data points available, while sensor-box 8 in Cham has 4401 hourly data points available. The difference in the length of the data set is largely due to the stability of the hardware: some sensor-boxes already required maintenance a few days after installation (e.g., Pilot Malters 2), while other boxes were continuously acquiring data without hardware issues over a longer period of time (e.g., Pilot Cham 8). The results of the statistical analysis of the field study are presented in Table 8. It can be seen that all data sets except for two ((S) and (U)) have a bias towards underestimating the actual PM concentration. In addition, all of the slopes are less than 1. While the validation measurements show a relatively good agreement with the reference measurements, the field study shows more varied results. The values of $R_{P}$ range from 0.21 (Pilot Malters 2) to 0.88 (Pilot Horw 3), with 67% of the values being larger than or equal to 0.5. The median value lies at $R_{P}=0.63$. The range of the $R_{S}$ values goes from 0.21 (Pilot Malters 2) to 0.91 (Pilot Horw 3), with 91% of the values being larger than or equal to 0.5. The median value lies at $R_{S}=0.73$. The correlations compared among the different pilot sites are also shown in Figure 12.

Table 9 presents the analysis of the statistical measures obtained across all 21 data sets. There, it can be seen that the average bias is an underestimation of 44%. The intercepts range from −2.69 μg/m${}^{3}$ to 5.60 μg/m${}^{3}$, while the Mean Absolute Error ranges from 2.49 μg/m${}^{3}$ to 12.52 μg/m${}^{3}$. Considering the magnitude of the bias and seeing that the average Spearman correlation is 0.61, it is assumed that the errors can largely be attributed to systematic errors of the sensor. This error could therefore be reduced with an appropriate calibration of the sensor (not part of this work).

In order to investigate the reason for the spread in $R_{P}$ values, selected pilots with different data patterns are studied more closely. In the following, two exemplary pilots from the data sets shown in Table 7 are presented in more detail. The selected pilots differ in the sense that each shows one of the following characteristics: either a high correlation between mobile pilot and in-luft data is observed most of the time or a high correlation between mobile pilot and in-luft data is observed at specific times, while a low correlation is observed in between.

Figure 13 shows the mean hourly values of PM${}_{10}$ data recorded with the mobile sensor-box and the stationary in-luft station, both located in Luzern. This is an example of a pilot showing a good correlation between the sensor-box data and in-luft station data most of the time. An offset between the two datasets can be observed, with the sensor-box data generally showing lower values than the in-luft data. This also becomes evident when looking at the scatter plot shown in Figure 14.

An example of a pilot with intermittent good correlation between sensor-box data and in-luft data is shown in Figure 15. The mobile sensor-box, as well as the stationary in-luft station, were located in Ebikon. It can be seen in the time-series graph that the sensor-box data do not follow the in-luft data as consistently as in the previously mentioned example of Luzern. Fluctuations in magnitude of the PM${}_{10}$ values can be observed: there are periods where the sensor-box data match closely the in-luft data and there are periods where the two data sets barely correlate. In order to better understand the reason for these fluctuations, the geo-location of the datapoints was considered. Analysis of this pattern showed that the periods of good correlation occur when the vehicle carrying the sensor-box is not located at the parking position (i.e., maintenance depot). On the contrary, when the vehicle is located at the parking position, the correlation is considerably worse. A more detailed analysis of this pattern is described in the subsequent section. The same pattern was also observed for other pilot locations such as, e.g., Hergiswil.

Looking at a shorter time period (e.g., two weeks) allows for a better understanding of the fluctuating PM${}_{10}$ values. Figure 16 shows the hourly mean PM${}_{10}$ data of the pilots located in Ebikon and Hergiswil from 12 December to 27 December 2021. Periods where the vehicle is thought to be in operation or parked outdoors are marked in red. In the periods in between, the vehicle was most likely located at the parking position indoors at the maintenance depot. There is a clear difference in magnitude of the values: during times when the vehicle was in operation, higher PM${}_{10}$ values were recorded. During the weekend (18–19 December), as well as during the night-time, when the vehicle was not in operation, the values remained low.

Based on above-mentioned findings, an additional filter based on the geo-location of the data points is tested on the data set. At both locations where this pattern occurs, the data within a radius of 150 m around the maintenance depot is removed. Therefore, only data when the vehicle is in operation remain. It was observed that such patterns occurred mainly for pilots where the parking position of the vehicle is located in a closed building, which also applies to the pilots in Ebikon and Hergiswil. The data set for Ebikon presented in Figure 15 is filtered by geo-location, removing all data points recorded in the vicinity of the maintenance depot. The resulting time-series compared to the in-luft data are shown in Figure 17, whereas the resulting scatter plot is presented in Figure 18. The number of hourly data points reduces from 768 to 129. The Pearson correlation increases from 0.67 to 0.81, whereas the Spearman’s correlation increases from 0.70 to 0.83. In addition, it can be seen from the time-seires graph that the sensor-box data follows the in-luft data more closely than was the case before the geo-location filter was applied.

### 3.4. Influence of Distance to Reference Station for Data Validation

It is expected that reference stations, which are located closer to a pilot and have similar topography and land use, show a better correlation with the collected pilot data in contrast to reference stations located further away from the pilot. For this purpose, the influence of the distance between the reference station and the mobile sensor-box is analyzed in this section. The pilot location Cham is compared to two different reference stations: the in-luft station Zug and the in-luft station Rigi. An overview of the geographical location of all three measurement locations is given in Figure 19. The two reference stations not only differ in terms of distance to the pilot location, but also in altitude and surrounding environment. The profiles of the two reference stations are described in Table 10. The in-luft station Zug is relatively close to the pilot location Cham and has a similar surrounding (urban; close to lake) as the area covered by the mobile sensor-box. The in-luft station Rigi is further away and has a different surrounding (rural; pre-alpine) compared to the pilot location.

As a first step of this analysis, the two stationary in-luft stations selected are compared to each other. Using hourly mean data sets obtained from the in-luft measurement stations, a comparison is made for the months of July and November 2021, allowing the investigation of two different seasons. During the month of July, the PM${}_{10}$ data sets for the reference stations Zug and Rigi look very similar, and a high Pearson correlation is observed ($R_{P}$ = 0.90). In autumn (i.e., November), however, the correlation is significantly lower ($R_{P}$ = 0.43). Time-series data for the two selected months are shown in Figure 20. Here, the difference between the data obtained in July and November can be seen: whereas for both months the values from the Rigi station are generally lower, the difference is larger in November than in July. In addition, the shapes of the profiles between the two stations show stronger differences in November than in July. Figure 21 shows the scatter plots of the PM${}_{10}$ data from the two reference stations. The linear correlation between the two data sets is higher in July than in November.

As a next step, the two stationary in-luft reference stations are compared to the mobile pilot in Cham. Based on the findings from the comparison of the two in-luft reference stations, it is expected that the comparison with the mobile pilots will show a similar picture. Therefore, a comparison is made between the data from the mobile pilot located in Cham and the two reference stations Zug and Rigi during the months of July and November. Figure 22 and Figure 23 show time-series graphs and scatter plots of the comparison with the in-luft station in Zug for the months of July and November. The in-luft station Zug is the closest reference station to the mobile pilot in Cham with a distance of approximately 5 km between in-luft station and pilot parking position. For both months, a relatively high correlation can be observed between the in-luft station Zug and the mobile pilot in Cham, whereas for July, it is higher ($R_{P}=0.82$) than in November ($R_{P}=0.69$). As previously shown in the comparison between the two in-luft stations Zug and Rigi, during the month of July, the correlation between the two stations was higher than in November. Therefore, it is possible that the lower correlation in November for the comparison between sensor-box Cham and in-luft station Zug stems from the same seasonal effect.

Figure 24 and Figure 25 show time-series graphs and scatter plots of the comparison between the mobile pilot in Cham and the in-luft station in Rigi for the months of July and November. The in-luft station Rigi is located further away from the mobile pilot in Cham with a distance of approximately 13 km between in-luft station and pilot parking position. In addition, it exhibits a different profile (altitude, surroundings, etc.) than the in-luft station in Zug (refer to Table 10). For the month of July, the correlation between sensor-box Cham and in-luft station Rigi is equally high as for the comparison with the in-luft station Zug, even though the in-luft station Rigi is located several kilometers further away from the mobile sensor-box and in a different geographical setting. During the month of November, however, the comparison shows a very low correlation. These results are in line with the findings presented previously when comparing the two in-luft stations Zug and Rigi to each other.

## 4. Conclusions

Mobile, low-cost sensor nodes offer a promising solution for obtaining a more extensive set of air quality data in communities at a much lower expense compared to existing stationary, high-precision reference stations. Such a mobile low-cost sensor-box was developed for the acquisition of air quality data in municipalities. Validation measurements were conducted where our sensor-boxes were placed directly adjacent to a reference station. Most studies about low-cost PM sensors found in the literature discuss calibration methods and results from post-calibration analysis. However, pre-processing stages are often not mentioned or not the focus of the study. Therefore, using the data from the validation measurement, in this study, several filtering methods were tested to remove outliers from the raw data sets before further analyzing the data. A suitable filtering method is applied in order to improve the data quality, without losing information about extreme values. After application of these filtering methods, linear correlation coefficients between 0.49 and 0.89 were achieved. Furthermore, the PM${}_{10}$ data of an 8-month field study carried out in Central Switzerland were analyzed and compared to measurements from stationary reference stations. As for the mobile field measurements, 67% of the sensor nodes achieved a linear correlation of 0.5 or higher, with a maximum of 0.88. Some sensor nodes showed a consistently good correlation with the reference station, even though there was a consistent bias towards the underestimation of the actual values observed in most of the sensor-box data sets. Other sensor nodes showed a good correlation during specific times only (e.g., for several hours during the day) and a low correlation for the remaining time. For these sensor nodes, an additional filter that removes measurements recorded at specific locations with atypical PM${}_{10}$ concentrations (such as a closed parking garage) was introduced. This yielded an improved correlation with the reference stations.

In addition, it was examined whether the profile of the reference stations (i.e., distance to mobile sensor-box and surroundings of the station) have an influence on the correlation between sensor-box data and reference data. This analysis was performed for one mobile pilot location. It was found that during summer months, the distance to the reference station, as well as the profile of the reference station, have less of an influence on the PM correlation than during the autumn or winter months. Therefore, it is recommended to use the closest and most similar reference station when comparing the mobile sensor-box data to reference data.

Future work could include the analysis of data acquired over several seasons (e.g., minimum 12 months). In addition, a calibration method for the mobile sensor nodes can be introduced based on the validation measurements and including the influence of humidity. For this purpose, it must be ensured that the reference station is exposed to the same conditions as the mobile sensor-box.

This study has shown methods of data treatment and the resulting statistical metrics without the application of a calibration, which provided important information about the use of low-cost PM sensing devices.

## Figures and Tables

**Figure 1 sensors-23-00794-f001:**
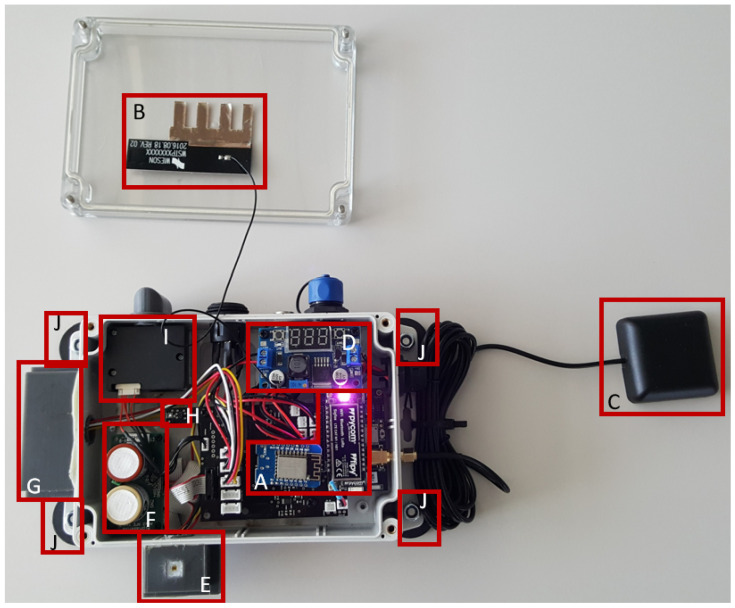
The sensor-box: (**A**) microcontrollers (FiPy; here ESP8266 instead of ESP32), (**B**) LTE antenna, (**C**) GPS antenna, (**D**) DC/DC converter, (**E**) Light sensor, (**F**) O${}_{3}$ and NO${}_{2}$ sensors, (**G**) Sound sensors, (**H**) Temperature/Humidity and TVOC/CO${}_{2}$ sensors, (**I**) PM sensor, (**J**) Magnets.

**Figure 2 sensors-23-00794-f002:**
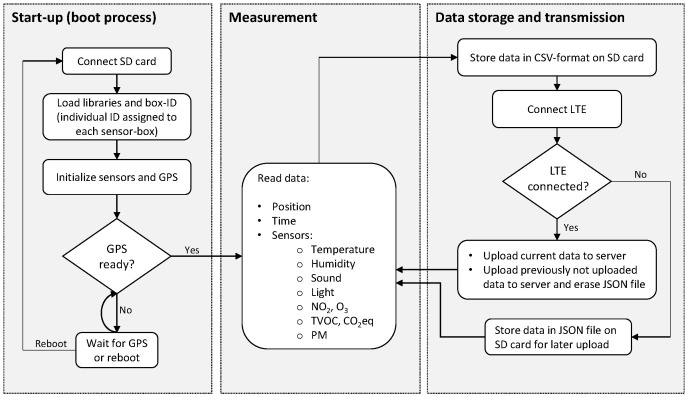
Software flow chart: sensor-box data acquisition and transmission cycles.

**Figure 3 sensors-23-00794-f003:**
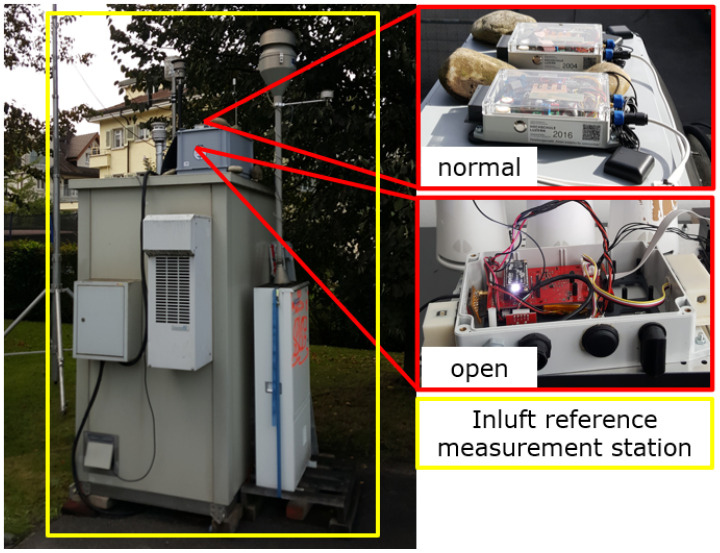
Validation campaign: sensor-boxes with IDs 1, 2 and 7 placed at the in-luft reference station in Stans.

**Figure 4 sensors-23-00794-f004:**
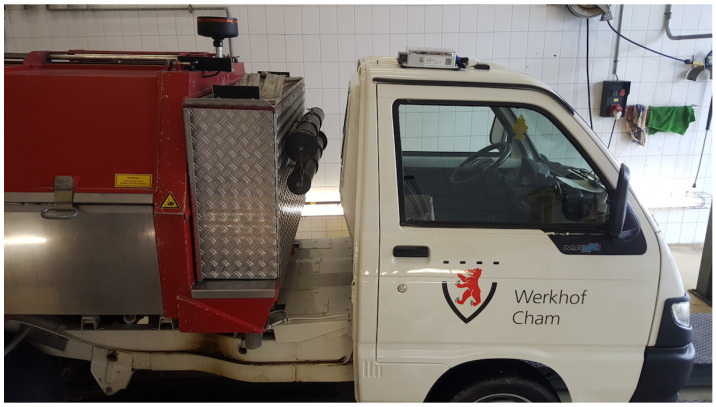
Sensor-box mounted on a utility vehicle from the municipality of Cham.

**Figure 5 sensors-23-00794-f005:**
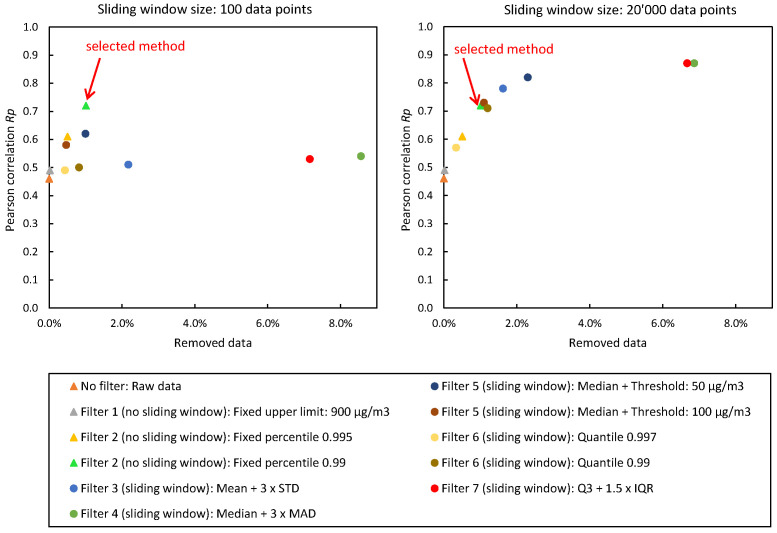
Validation with reference station: evaluation of filtering methods using PM${}_{10}$ concentration measurements recorded with stationary sensor-box 2 between 17 November 2021 and 31 December 2021 in Stans, Nidwalden. Displayed is the Pearson correlation coefficient of in-luft and measurement data with different thresholds for the data selection.

**Figure 6 sensors-23-00794-f006:**
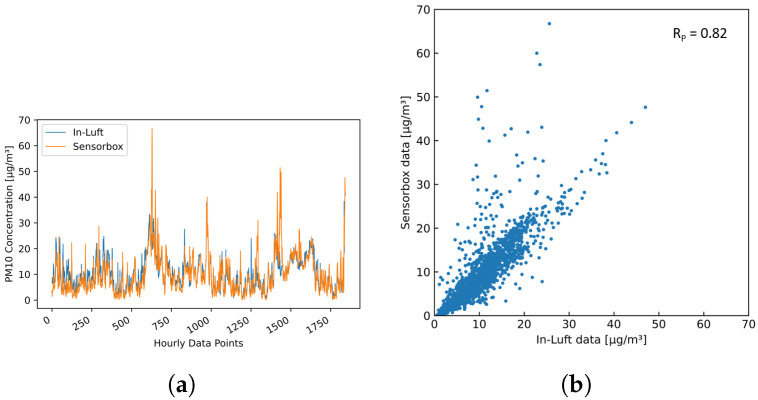
PM${}_{10}$ hourly mean data recorded with sensor-box 7 located in Stans in the period from 15 October 2021 to 31 December 2021, compared to hourly mean data recorded at the in-luft station located in Stans. Fixed-percentile (Filter 2, 99.0%) applied to sensor-box data. N=1843, $R_{P}=0.82$, $R_{S}=0.90$ (**a**) time series; (**b**) scatter plot.

**Figure 7 sensors-23-00794-f007:**
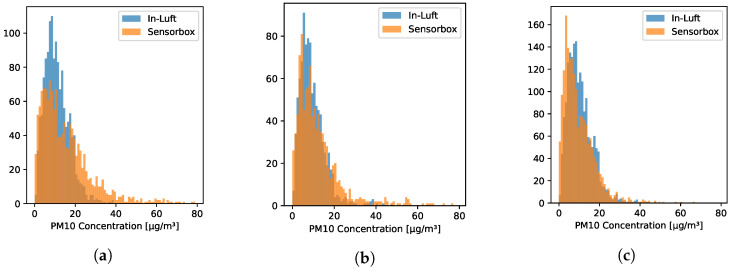
Distribution of hourly mean values of PM${}_{10}$ concentration recorded in Stans, Nidwalden. (**a**) Sensor-box 1 recorded between 15 October 2021 and 23 December 2021; (**b**) Sensor-box 2 recorded between 17 November 2021 and 31 December 2021; (**c**) Sensor-box 7 recorded between 15 October 2021 and 31 December 2021.

**Figure 8 sensors-23-00794-f008:**
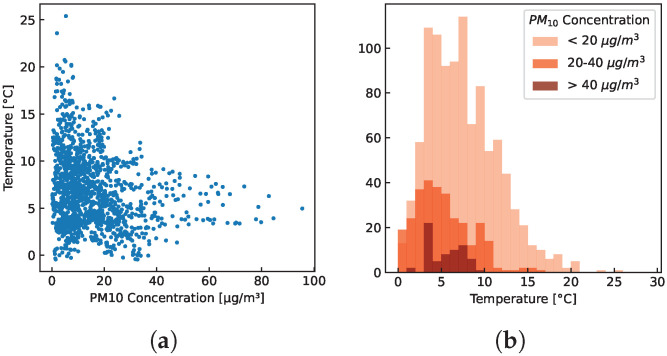
Hourly mean values recorded with sensor-box 1 between 15 October 2021 and 23 December 2021 in Stans, Nidwalden. (**a**) PM${}_{10}$ concentration vs. temperature; (**b**) Distribution of three different PM${}_{10}$ concentration ranges.

**Figure 9 sensors-23-00794-f009:**
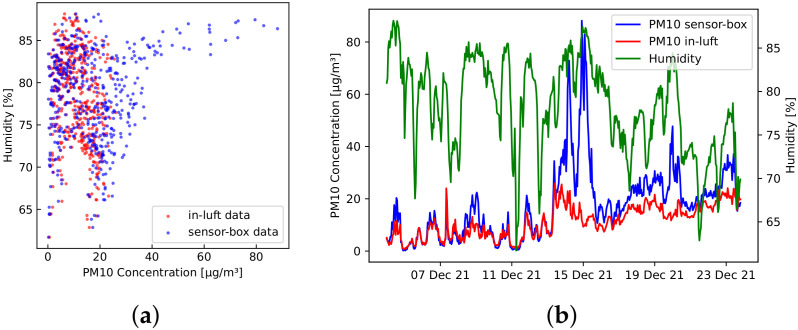
Hourly mean values recorded with sensor-box 1 between 4 December 2021 and 24 December 2021 in Stans, Nidwalden compared to in-luft data measured in the same time-interval. (**a**) PM${}_{10}$ concentration vs. humidity; (**b**) PM${}_{10}$ and humidity time-series data.

**Figure 10 sensors-23-00794-f010:**
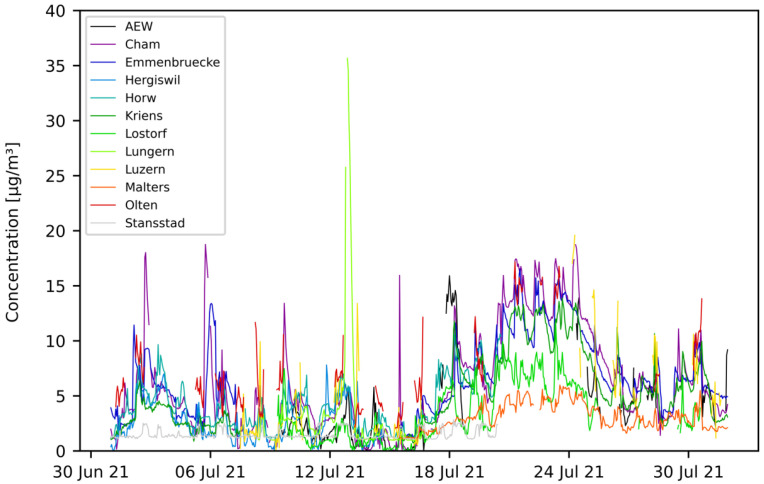
PM${}_{10}$ hourly mean data recorded with mobile pilot sensor-boxes in the period from 1 July 2021 to 30 July 2021.

**Figure 11 sensors-23-00794-f011:**
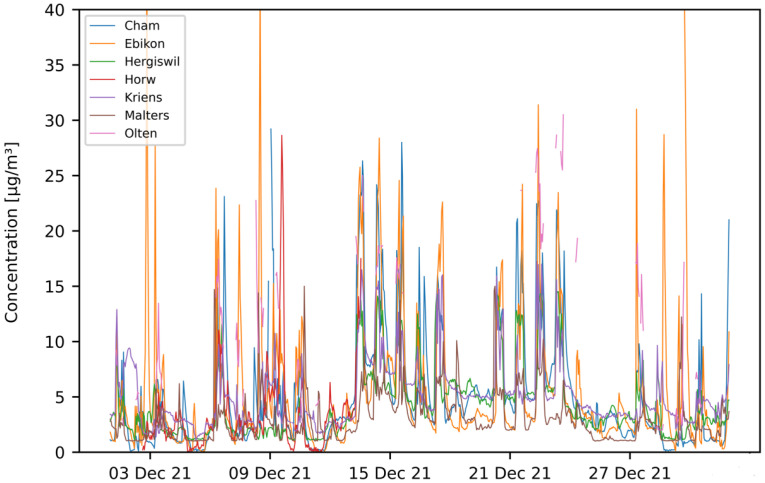
PM${}_{10}$ hourly mean data recorded with mobile pilot sensor-boxes in the period from 1 December 2021 to 31 December 2021.

**Figure 12 sensors-23-00794-f012:**
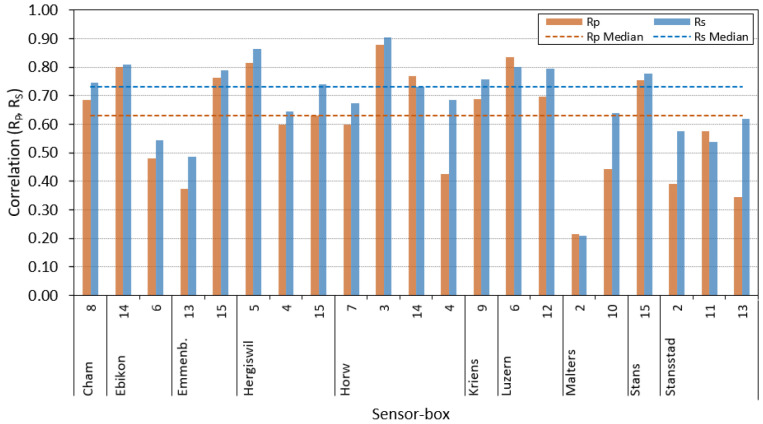
Pearson correlation $R_{P}$ and Spearman correlation $R_{S}$ for mean hourly PM${}_{10}$ data between sensor-box and in-luft stations compared among different pilot sites. Evaluation of data collected between May and December 2021. Fixed percentile filtering method (99.0%) is applied to the raw sensor-box data.

**Figure 13 sensors-23-00794-f013:**
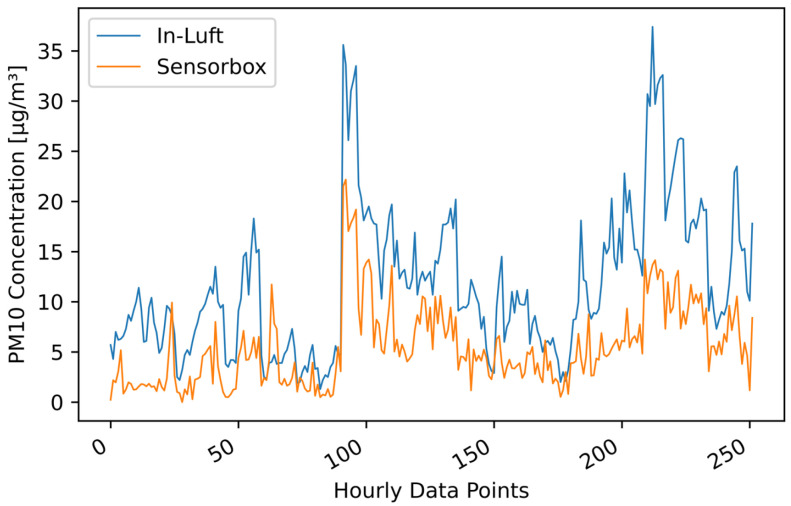
PM${}_{10}$ hourly mean data recorded with mobile pilot sensor-box 6 located in Luzern in the period from 7 July 2021 to 2 September 2021, compared to hourly mean data recorded at the in-luft station located in Luzern. Data gaps are removed from the graph. Number of mean hourly values: 252; Resulting correlations: $R_{P}=0.83$, $R_{S}=0.80$.

**Figure 14 sensors-23-00794-f014:**
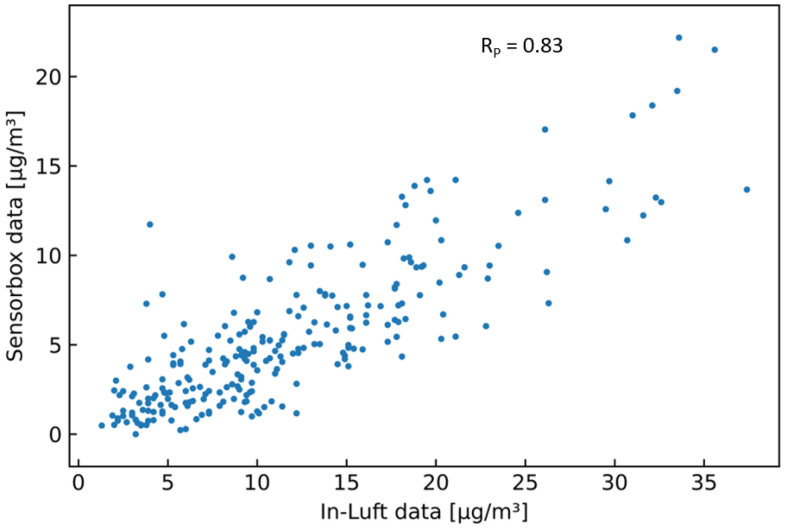
PM${}_{10}$ hourly mean data recorded with mobile pilot sensor-box 6 located in Luzern in the period from 7 July 2021 to 2 September 2021, compared to hourly mean data recorded at the in-luft station located in Luzern. Number of mean hourly values: 252; Resulting correlations: $R_{P}=0.83$, $R_{S}=0.80$.

**Figure 15 sensors-23-00794-f015:**
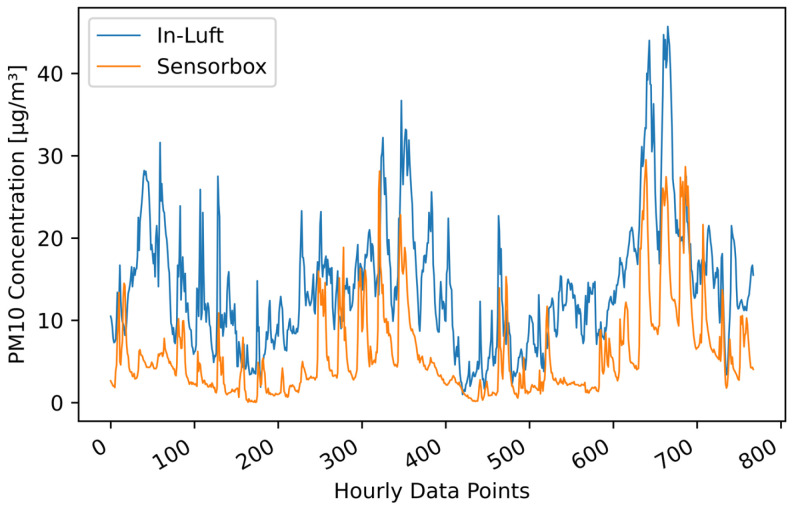
PM${}_{10}$ hourly mean data recorded with mobile pilot sensor-box 6 located in Ebikon in the period from 15 October 2021 to 15 November 2021, compared to hourly mean data recorded at the in-luft station located in Ebikon, Sedel. Data gaps are removed from graph. Number of mean hourly values: 768; Resulting correlations: $R_{P}=0.67$, $R_{S}=0.70$.

**Figure 16 sensors-23-00794-f016:**
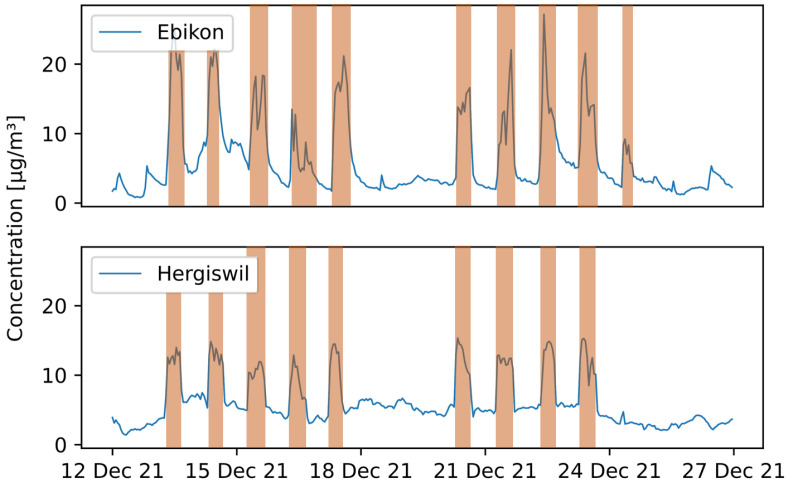
PM${}_{10}$ hourly mean data recorded with mobile pilot sensor-boxes located in Ebikon and Hergiswil in the period from 12 December 2021 to 27 December 2021. Periods where the vehicle is in operation are marked in red.

**Figure 17 sensors-23-00794-f017:**
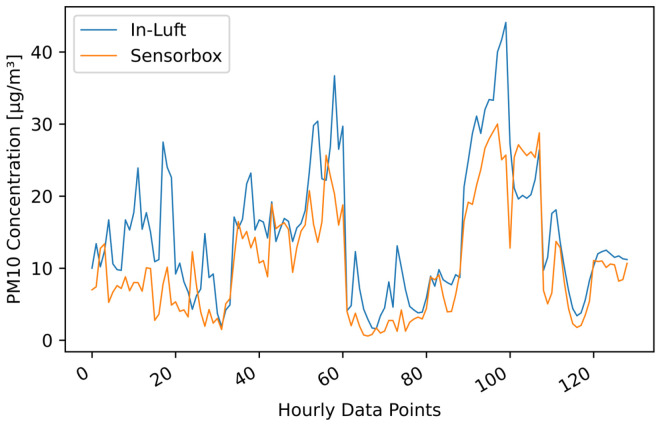
PM${}_{10}$ hourly mean data recorded with mobile pilot sensor-box 6 located in Ebikon in the period from 15 October 2021 to 15 November 2021 compared to hourly mean data recorded at the in-luft station located in Ebikon, Sedel. Data gaps are removed from graph. Data recorded within a radius of 150 m around the maintenance depot Ebikon are removed. Number of mean hourly values: N = 129; Resulting correlations: $R_{P}=0.81$, $R_{S}=0.83$.

**Figure 18 sensors-23-00794-f018:**
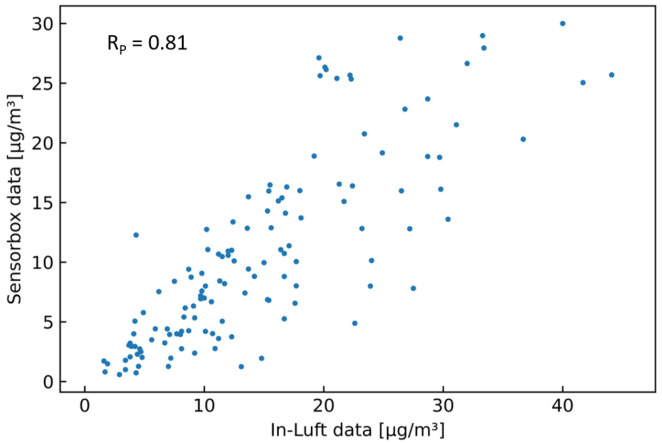
PM${}_{10}$ hourly mean data recorded with mobile pilot sensor-box 6 located in Ebikon in the period from 15 October 2021 to 15 November 2021 compared to hourly mean data recorded at the in-luft station located in Ebikon, Sedel. Data recorded within a radius of 150 m around the maintenance depot Ebikon are removed. Number of mean hourly values: N = 129; Resulting correlations: $R_{P}=0.81$, $R_{S}=0.83$.

**Figure 19 sensors-23-00794-f019:**
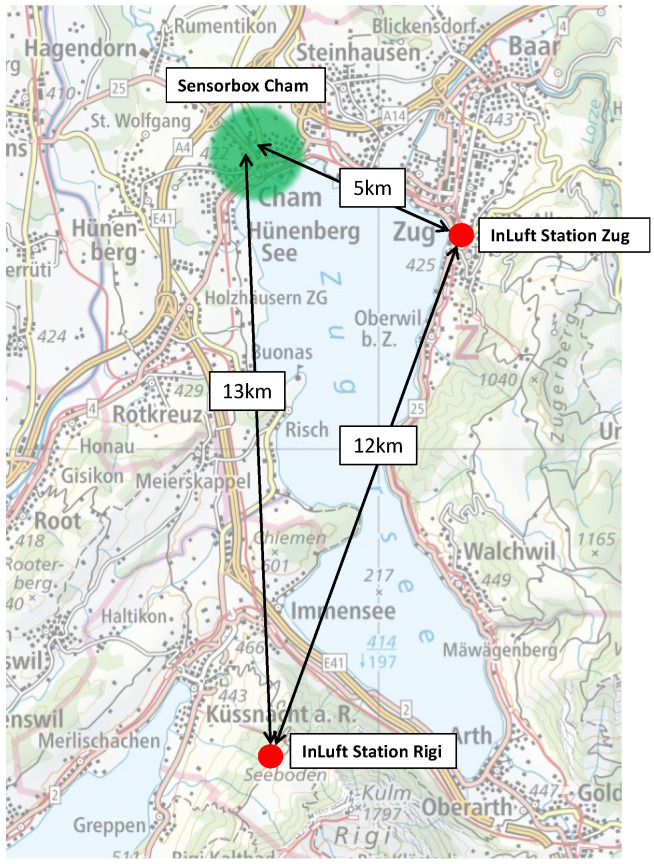
Geographical location of pilot in Cham and in-luft reference stations Zug and Rigi. (Map source: Federal Office of Topography).

**Figure 20 sensors-23-00794-f020:**
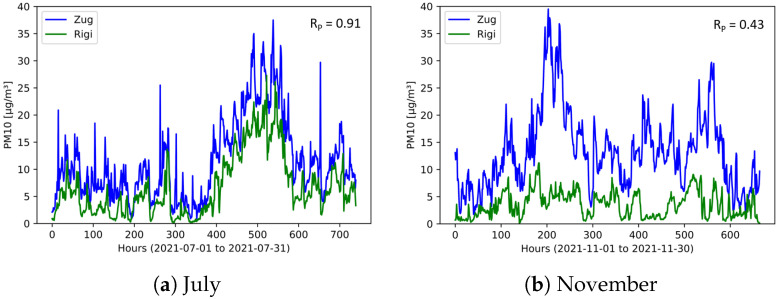
Hourly mean PM${}_{10}$ data recorded at in-luft stations Zug and Rigi. (**a**) N=740, $R_{P}=0.91$; (**b**) N=665, $R_{P}=0.43$.

**Figure 21 sensors-23-00794-f021:**
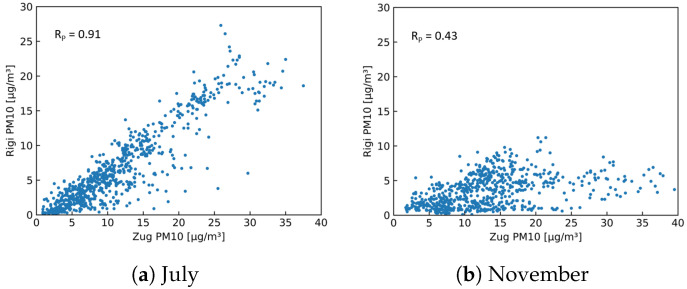
Hourly mean PM${}_{10}$ data measured at in-luft stations Zug and Rigi. (**a**) N=740, $R_{P}=0.91$; (**b**) N=665, $R_{P}=0.43$.

**Figure 22 sensors-23-00794-f022:**
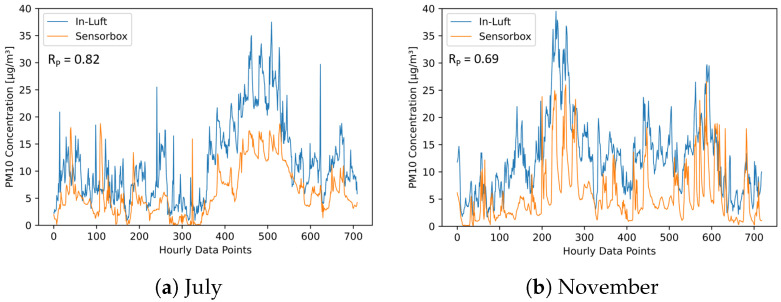
PM${}_{10}$ hourly mean data recorded with mobile pilot sensor-box 8 located in Cham compared to hourly mean data recorded at the in-luft station located in Zug, Postplatz. Data gaps are removed from the graph. (**a**) N=710, $R_{P}=0.82$, $R_{S}=0.81$; (**b**) N=719, $R_{P}=0.69$, $R_{S}=0.70$.

**Figure 23 sensors-23-00794-f023:**
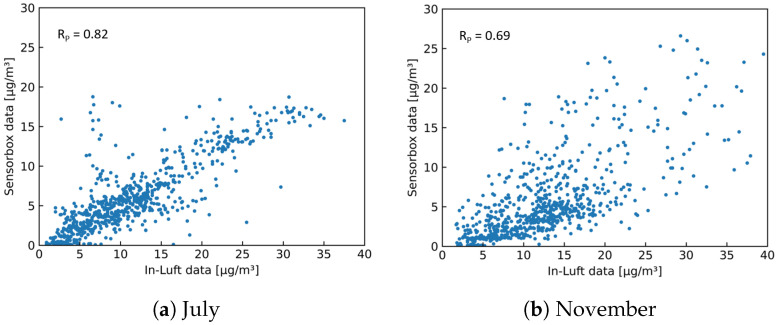
PM${}_{10}$ hourly mean data recorded with mobile pilot sensor-box 8 located in Cham compared to hourly mean data recorded at the in-luft station located in Zug, Postplatz. (**a**) N=710, $R_{P}=0.82$, $R_{S}=0.81$; (**b**) N=719, $R_{P}=0.69$, $R_{S}=0.70$.

**Figure 24 sensors-23-00794-f024:**
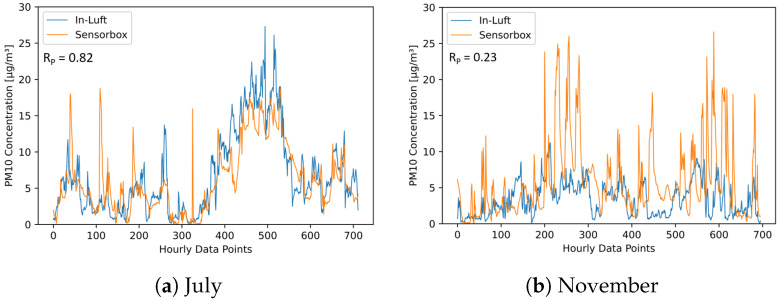
PM${}_{10}$ hourly mean data recorded with mobile pilot sensor-box 8 located in Cham compared to hourly mean data recorded at the in-luft station located in Rigi, Seebodenalp. Data gaps are removed from the graph. (**a**) N=712, $R_{P}=0.82$, $R_{S}=0.81$; (**b**) N=696, $R_{P}=0.23$, $R_{S}=0.32$.

**Figure 25 sensors-23-00794-f025:**
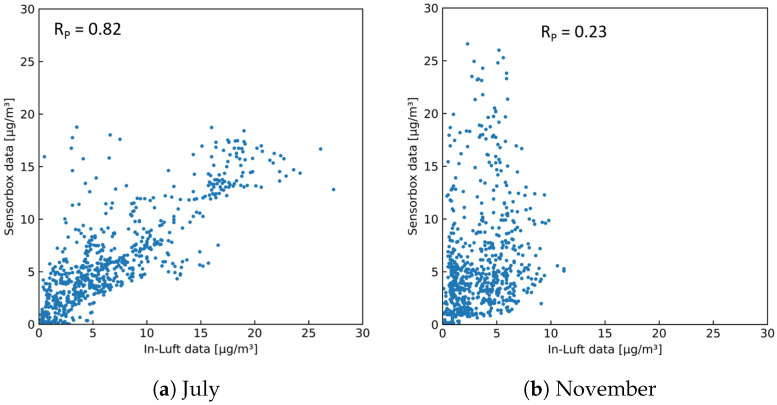
PM${}_{10}$ hourly mean data recorded with mobile pilot sensor-box 8 located in Cham compared to hourly mean data recorded at the in-luft station located in Rigi, Seebodenalp. (**a**) N=712, $R_{P}=0.82$, $R_{S}=0.81$; (**b**) N=696, $R_{P}=0.23$, $R_{S}=0.32$.

**Table 1 sensors-23-00794-t001:** Overview of experiments and field tests comparing particulate matter measurements from low-cost sensors to reference instruments.

Location	Experiment Setup and Main Conclusions
	Sensor type; low-cost sensor make; position relative to reference station; environment; results
Aveiro, Portugal [16]	Optical; Shinyei PPD42, Shinyei PPD20V, others; side-by-side; outdoors
	PM_10_: $r^{2}$ (0.13–0.36); PM_2.5_: $r^{2}$ (0.07–0.27)
Oslo, Norway [17]	Optical; AQMesh units; side-by-side; outdoors (dense traffic vs. calm traffic)
	PM_10_: $r^{2}=0.53$ (dense traffic), $r^{2}=0.68$ (calm traffic); PM_2.5_: $r^{2}=0.40$ (dense traffic), $r^{2}=0.84$ (calm traffic); Average match score for PM_10_ 0.91, PM_2.5_ 0.48
Ispra and Brindisi, Italy [22]	Optical; Shinyei PPD20V; side-by-side; outdoors. Period December 2013–March 2014, 1 sample per minute, two locations one rural setting and one industrial site
	Accuracy of the calibrated optical particle sensor has been calculated as mean error and max error compared to the PM${}_{10}$ referenced analyzer. They are estimated at 9.0 μg/m^3^ and 41.7 μg/m^3^
Bari, Italy [15]	Optical; Shinyei PPD20V; various locations indoors and outdoors; 11 nodes (10 stationary and 1 mobile mounted on public bus); results are compared to closest air quality monitoring station.
	MAE ^1^: 5.6 μg/m^3^, Accuracy ^2^ in node 1, 2, and 3 is 24.8%, 21.6%, and 20.5%)
Helsinki, Finland	Ref. [23]: various types; make not specified; outdoors in 3 different environments (industry with congested traffic; residential with low traffic; mixed residential and university); 100 mobile sensors; 12 fixed sensors; additional sensors side-by-side with reference stations; absolute error after calibration with data in the vicinity of reference stations: PM_10_ (2.88–17.84 μg/m^3^); PM_2.5_ (1.38–9.09 μg/m^3^)
	Ref. [20]: Optical; Panasonic; personal exposure, indoor and outdoor; comparison to reference station 7 km away: R = 0.5
Seoul, South Korea	Ref. [18]: Optical; PMS7003 (Plantower Inc.); outdoors; side-by-side; after calibration (combined linear and non-linear): PM_2.5_ RMSE = 4.70 μg/m^3^, $R^{2}$ = 0.89
	Ref. [19]: Optical; Sensirion SPS30; side-by-side; PM_2.5_ after calibration (neural network): $R^{2}$ (0.59–0.93)
Badajoz, Spain [21]	Optical; Alphasense OPC-N3; side-by-side, portable sensor-box validation with a mobile reference measurement station, PM${}_{2.5}$, PM${}_{10}$ at 3 s resolution, averaged over 10 min and 1 h,
	PM_10_: $R^{2}$ (0.48–0.78); PM_2.5_: $R^{2}$ (0.22–0.64)

^1^ Mean Absolute Error, ^2^ Defined as percentage ratio MAE divided by reference data mean.

**Table 2 sensors-23-00794-t002:** Specifications of the Cubic PM3015SN particulate matter sensor [24] ^1^.

Specifications	Value
Operating principle	Laser scattering
Measured particle size range	0.3–10 μm
Measurement range	0–5000 μg/m${}^{3}$
Resolution	1 μg/m${}^{3}$
Working condition	−15 to $70{\;}^{\circ }$C, 0–95% RH
Measurement accuracy PM_1.0_ and PM_2.5_	0–100 μg/m${}^{3}$, ±5 μg/m${}^{3}$
	101–1000 μg/m${}^{3}$, ±15% of reading
	Condition: 25 ± 2 ${}^{\circ }$C, 50 ± 10% RH
Measurement accuracy PM_10_	0–100 μg/m${}^{3}$ ± 30 μg/m${}^{3}$
	101–1000 μg/m${}^{3}$ ± 30% of reading
	Condition: 25 ± 2 ${}^{\circ }$C, 50 ± 10% RH
Response time	1 s
Time to first reading	≤8 s

^1^ Data sheet not available online, contact manufacturer.

**Table 3 sensors-23-00794-t003:** Technical specifications of the reference PM measurement station, Fidas200 [3,27].

Specifications	Value
Operating principle	Laser scattering
Particle range	0.18–18 μm
Resolution	0.1 μm/m${}^{3}$
Working condition	5 to $40{\;}^{\circ }$C
Measurement accuracy PM_2.5_	9.7%
Measurement accuracy PM_10_	7.5 %
Response time	<2 s

**Table 4 sensors-23-00794-t004:** Pilot overview of the field test measurement campaign.

Community	Start	End	Duration (Months)
Hergiswil	April 2021	April 2022	12
Rheinfelden (AEW)	May 2021	July 2021	3
Stansstad	May 2021	November 2021	6
Lostorf	May 2021	April 2022	11
Stans	May 2021	November 2021	6
Horw	May 2021	April 2022	11
Lungern	June 2021	April 2022	10
Kriens	June 2021	April 2022	10
Olten	June 2021	April 2022	10
Malters	June 2021	March 2022	9
Cham	June 2021	April 2022	10
Emmenbruecke	June 2021	April 2022	10
Luzern	July 2021	April 2022	9
Ebikon	September 2021	April 2022	7

**Table 5 sensors-23-00794-t005:** Stationary sensor-boxes in Stans, Nidwalden.

Sensor-Box ID	Start–End	Duration	Data Points
1	15 October–23 December 2021	2 months	107,338
2	17 November–31 December 2021	1.5 months	71,149
7	15 October–31 December 2021	2.5 months	134,768

**Table 6 sensors-23-00794-t006:** Results of statistical analysis of stationary senor-box measurements in Stans, Nidwalden. Measurements recorded between 15 October 2021 and 31 December 2021. Fixed percentile filtering method (99.0%) is applied to the raw data. No further calibration applied.

Sensor-Box ID	MAE (μg/m${}^{3}$)	RMSE (μg/m${}^{3}$)	Slope	Intercept (μg/m${}^{3}$)	$R_{P}$	$R_{S}$	Bias (%)
1	5.44	10.38	1.60	−2.14	0.74	0.88	30.30
2	3.70	8.23	1.28	0.06	0.72	0.87	27.40
7	2.52	4.38	1.01	−0.79	0.82	0.90	−10.73

**Table 7 sensors-23-00794-t007:** Overview of usable data sets collected between May and December 2021 in Central Switzerland.

Data Set	Pilot	Sensor-Box	In-Luft Station	Distance	Hourly Data Points
(A)	Cham	8	Zug	5 km	4401
(B)	Ebikon	14	Ebikon	3 km	240
(C)		6			1878
(D)	Emmenbruecke	13	Ebikon	1 km	166
(E)		15			1127
(F)	Hergiswil	5	Stans	5 km	2125
(G)		4			477
(H)		15			1210
(I)	Horw	7	Luzern	4 km	118
(J)		3			1226
(K)		14			155
(L)		4			337
(M)	Kriens	9	Luzern	3 km	4073
(N)	Luzern	6	Luzern	0 km	252
(O)		12			401
(P)	Malters	2	Luzern	9 km	75
(Q)		10			3305
(R)	Stans	15	Stans	2 km	1223
(S)	Stansstad	2	Stans	3 km	784
(T)		11			837
(U)		13			2214

**Table 8 sensors-23-00794-t008:** Results of statistical analysis of pilot data against in-luft data for data collected between May and December 2021. Fixed percentile filtering method (99.0%) is applied to the raw data. No further calibration applied.

Data Set	MAE (μg/m${}^{3}$)	RMSE (μg/m${}^{3}$)	Slope	Intercept (μg/m${}^{3}$)	$R_{P}$	$R_{S}$	Bias (%)
(A)	6.71	8.39	0.44	0.59	0.69	0.75	−49.71
(B)	6.72	7.58	0.53	−0.51	0.80	0.81	−47.53
(C)	8.36	10.32	0.36	1.09	0.48	0.54	−44.30
(D)	3.47	4.28	0.29	2.85	0.37	0.48	−26.34
(E)	6.95	9.45	0.28	2.78	0.76	0.79	−41.03
(F)	4.89	6.19	0.47	−0.07	0.82	0.86	−57.16
(G)	7.26	8.68	0.44	−1.25	0.60	0.65	−71.36
(H)	5.86	7.59	0.34	1.12	0.63	0.74	−50.16
(I)	3.36	3.88	0.36	−0.07	0.60	0.67	−66.07
(J)	6.89	9.06	0.32	1.13	0.88	0.91	−54.90
(K)	9.18	10.17	0.59	−2.69	0.77	0.73	−61.31
(L)	7.27	8.93	0.25	1.13	0.43	0.69	−62.07
(M)	10.02	12.58	0.23	1.79	0.69	0.76	−58.40
(N)	6.47	7.79	0.46	0.02	0.83	0.80	−52.12
(O)	8.14	10.43	0.53	0.60	0.70	0.79	−45.62
(P)	2.49	3.18	0.09	5.60	0.21	0.21	−1.20
(Q)	12.52	15.00	0.09	1.48	0.44	0.64	−76.81
(R)	4.93	6.05	0.36	1.22	0.75	0.78	−42.14
(S)	2.67	4.27	0.45	3.88	0.39	0.58	47.14
(T)	7.98	11.03	0.04	1.33	0.58	0.54	−72.49
(U)	6.08	9.03	0.52	4.52	0.34	0.62	10.93

**Table 9 sensors-23-00794-t009:** Analysis of statistical metrics across all 21 pilot data sets.

	Mean	Min.	Max.	SD	Variance	CV (%)
$R_{P}$	0.61	0.21	0.88	0.18	0.03	30.09
$R_{S}$	0.68	0.21	0.91	0.15	0.02	22.23
Slope	0.35	0.04	0.59	0.15	0.02	42.41
Intercept	1.26	−2.69	5.60	1.86	3.46	147.14
Bias	−43.94	−76.82	47.14	29.18	851.48	−66.42
MAE	6.58	2.49	12.52	2.40	5.77	36.49
RMSE	8.28	3.18	15.00	2.88	8.32	34.84

**Table 10 sensors-23-00794-t010:** Location profile of in-luft reference stations [47].

Specification	Zug	Rigi
Geography	midlands	pre-alpine
Location	city center; close to lake	rural area; in open field close to forest
Altitude	420 m.a.s.l.	1031 m.a.s.l.
Settlement size	26,000	n/a
Distance to road	24 m	n/a

## Data Availability

Data available on request.

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
