# Peer review of "Low-Cost Sensor Node for Air Quality Monitoring: Field Tests and Validation of Particulate Matter Measurements"

_sensors, 2023, doi:10.3390/s23020794_

Round 1
Reviewer 1 Report
The authors developed and evaluated a low-cost air quality monitor in environmental settings. The low-cost monitors were mounted on mobile vehicles and set side-by-side with reference instruments. The data was then compared with the reference instrument during a year-long study. The work is intriguing, but the manuscript must be improved before publication.
Major Comments
Section 3.1 through 3.3 does not have enough data analysis, and the authors must add accuracy, bias, and precision calculations before this manuscript can be accepted for publication. Correlation is NOT enough; add slope, intercept and bias. In addition to adding the humidity effects on the low-cost sensors by comparing with the reference instrument.
The manuscript has a fundamental problem citing papers. The authors are not using the correct method for in-text citations. When you want to call a paper in the sentence directly, you DO NOT call the paper by calling its reference number.
For example
A recent study conducted by [3] shows …. [This is NOT correct]
A recent study conducted by Chen et al. (2021) shows …. [This is correct]
Now this is different than:
Recent studies conducted in Europe show that ….. [3]. Here you are not trying to call the paper directly. You are citing the sentence.
I am going to correct only TWO sentences, then I the authors should correct the remaining text before this manuscript can be accepted for publication.
A) The sentence on lines 19-22
The recent WHO global air quality guideline [1] recommends to set interim targets and progress towards lower maximum levels of particulate matter (e.g. PM2.5 and PM10), ozone, nitrogen, sulfur dioxide (SO2), and carbon monoxide.
Move the citation to the end of the sentence:
The recent WHO global air quality guideline recommends to set interim targets and progress towards lower maximum levels of particulate matter (e.g. PM2.5 and PM10), ozone, nitrogen, sulfur dioxide (SO2), and carbon monoxide [1].
B) The sentence on lines 23-25
Recent European exposure assessment studies, such as [3] and [4], report potentially increased mortality given the exposure to several compounds that are found in dust particles.
Here is a suggestion:
Recently, Chen et al. (2021) and Rodopoulou et al. (2021) conducted aerosol/air quality (?) exposure assessment studies in Europe and reported potentially increased mortality given the exposure to several compounds that are found in dust particles.
Please correct the remaining text in the manuscript. I will NOT refer to these problems below.
Detailed Comments:
1. Lines 17-19: Please add citation(s).
2. Lines 27-30: Please add citation(s) to support this statement and break up the sentence. It is long.
3. Lines 31-36: Please add citation(s).
4. Line 45: What is a “very good correlation”? Add values here and in the next sentences.
5. Paragraphs 40-72:
a. Need some background: Why are low-cost sensors used instead of reference instruments? Cost
b. Are these sensors OPCs or photometers? Explain the difference if both
c. Please add the names of the low-cost sensors used. Please explain what calibration means. Linear calibration?
6. Paragraphs 74-92: the objective should be only one paragraph NOT TWO. Remove one and include a paragraph about air quality measurements (OPCs) and how they work, as mentioned in 5. Also, mention that gravimetric measurements are the standard reference methods, and that optical methods are alternatives that are less accurate.
7. Line 156: The authors must mention that the device, although expensive, does not use a filter to correct the measurements. However, please mention if the device has a built-in heater to reduce humidity. Does it? If it does, why is that important?
8. Lines 188-191: Footnotes 4 and 5 should be moved to Introduction and provide a better background for the manuscript.
9. Where is the data analysis sub-section in the Methods section? How did the authors validate the data? It must be explained in the Methods section before presenting the results.
10. Line 247: Please change 3. Results to 3. Results and Discussion
11. Lines 249-252: Please move to the Methods section
12. Section 3.1 through 3.3 lacks accuracy and bias analysis between the low-cost sensors and the in-luft reference instrument, in addition to precision calculation (among sensors). Please add this data analysis. Please read the following paper for details on how this can be done (Laboratory and Field Evaluations of the GeoAir2 Air Quality Monitor for Use in Indoor Environments - Aerosol and Air Quality Research (aaqr.org))
13. Figures 8b and 9b are not useful, and the authors must add the in-luft measurements, so we can see the TRU effects of humidity on the low-cost sensor measurements. Then, the authors should discuss these results and compare them with other studies.
14. Lines 352-355: Please reformat and remove these points and rewrite them in the text. This is only used for reports for manuscripts. We do not use numbering or bullet points. Do the same for the remaining manuscript, converting numbering sentences to normal text.
Reviewer 2 Report
Low-Cost Sensor Node for Air Quality Monitoring: Field Tests and Validation of Particulate Matter Measurements
The manuscript is well-written and easy to follow.
Some points need to be known.
· How the sensor node is low cost as authors are using a variety of sensors (NO2, O3, TVOC, CO2, PM1, PM2.5, PM10, temperature, humidity, ambient sound, and ambient light) to measure the air quality and environmental conditions.
· What is the approximate cost of the proposed sensor node?
· It will be good to compare the proposed sensor node with other existing nodes, especially in terms of cost.
· Line 107: Why three microcontrollers are used as one is enough to do all the necessary tasks?
· Can the author please mention the Three microcontrollers (FiPy, 107 PyTrack, and ESP32) in Figure 1?
· As authors are using 3 microcontrollers, how it will be low cost? Also, authors are using LoRa devices, GSM devices,s, etc. Please clear.
· It will be good to highlight the key contributions of the proposed work (in the abstract as well as in the conclusions).
· Moderate English changes are required throughout the manuscript.
Round 2
Reviewer 1 Report
The authors have addressed my comments and accept the manuscript for publication
Reviewer 2 Report
Sensors (ISSN 1424-8220)
sensors-2059753
Title: Low-Cost Sensor Node for Air Quality Monitoring: Field Tests and Validation of Particulate Matter Measurements
Thank you for allowing me to revise resubmitted manuscript titled " Low-Cost Sensor Node for Air Quality Monitoring: Field Tests and Validation of Particulate Matter Measurements." All my previous comments have been adequately addressed, and I believe the submitted manuscript and presented work are suitable for publishing in the Sensors (ISSN 1424-8220). I suggest accepting the manuscript.